



**Interpreting Summertime Hourly Variation of NO₂ Columns with Implications for**
**Geostationary Satellite Applications**
Deepangsu Chatterjee[1], Randall V. Martin[1], Chi Li[1], Dandan Zhang[1], Haihui Zhu[1], Daven K. Henze[2],
James H. Crawford[3], Ronald C. Cohen[4,5], Lok N. Lamsal[6], Alexander M. Cede[6]
[1]Department of Energy, Environmental & Chemical Engineering, Washington University in St. Louis, St.
Louis, MO, USA
[2]Department of Mechanical Engineering, University of Colorado, Boulder, CO, USA
[3]NASA Langley Research Center, Hampton, VA, USA
[4]Department of Chemistry, University of California, Berkeley, Berkeley, CA, USA
[5]Department of Earth and Planetary Science, University of California, Berkeley, Berkeley, CA, USA
[6]NASA Goddard Space Flight Center, Greenbelt, MD 20771, USA
Correspondence: Deepangsu Chatterjee (deepangsuchatterjee@wustl.edu)
Abstract
Accurate representation of the hourly variation of the NO₂ column-to-surface relationship is needed to
interpret geostationary constellation observations of tropospheric NO₂ columns. Prior work has revealed
inconsistency in the hourly variation in NO₂ columns and surface concentrations. In this study, we use the
high-performance configuration of the GEOS-Chem model (GCHP) to interpret the daytime hourly
variation in NO₂ total columns and surface concentrations during summer. We use summer-time Pandora
sun photometers and aircraft measurements during the Deriving Information on Surface Conditions from
Column and Vertically Resolved Observations Relevant to Air Quality (DISCOVER-AQ) field campaign
over Maryland, Texas, and Colorado as well as 50 sites (31: contiguous USA, 10: Europe, 9: Asia) from
the Pandonia Global Network (PGN). We correct the Pandora columns for 1) hourly variation in the column
effective temperature driven by the fractional boundary layer contribution to the total column, and 2) change
in local solar time along the line-of-sight of the Pandora instrument. The corrected Pandora observations
are increased by about 5-6 × 10¹⁴ molecules cm⁻² at 9 AM and 6 PM across all Pandora sites. We conduct
fine resolution (~12 km) simulations over the contiguous US, Europe, and East Asia using the stretched
grid capability of GCHP. We also examine the effect of planetary boundary layer height (PBLH) corrections
on the total columns. We first evaluate the GCHP simulated absolute NO₂ concentration with Pandora and
aircraft observations. We find that fine resolution simulations at 12 km compared with moderate resolution
~55 km reduce the Normalized Bias (NB) versus Pandora total columns (19% to 10%) and versus aircraft
measurements (25% to 13%) over Maryland, Texas, and Colorado. Fine resolution simulations at 12 km
compared with moderate resolution at 55 km reduce the NB versus Pandora total columns over the eastern
US (17% to 9%), western US (22% to 14%), Europe (24% to 15%), and Asia (29% to 21%). We next use
the 12 km simulation to examine the hourly variation in the NO₂ column and surface concentrations. We
explain the weaker hourly variation in NO₂ columns than at the surface as a function of 1) hourly variation
in the column effective temperature, 2) hourly variation in the local solar time along the Pandora line-of-
sight and 3) the integral of weakly connected layers; with the lowest 500 m exhibiting greater NO₂
concentrations in morning and evening than midday, while the residual column above 500 m dominates the
total column with weaker variability.




## 1    Introduction


Nitrogen oxides ($NO_x \equiv NO + NO_2$) affect air quality and human health directly by

contributing to premature mortality (Burnett et al., 2004; Tao et al., 2012) and asthma for children
and adults (Anenberg et al., 2018), and indirectly by acting as precursors for tropospheric ozone
($O_3$) formation (Jacob et al., 1996), and nitrate aerosols (Bauer et al., 2007). Significant gaps in
ground-based monitoring of surface $NO_2$ concentrations and pronounced $NO_2$ heterogeneity
inhibit exposure assessment. To fill in knowledge of $NO_2$ exposures across a greater fraction of
the human population, satellite remote sensing offers the potential for spatially comprehensive
measurements. Major advances in satellite remote sensing from sun-synchronous low earth orbit
(LEO) have achieved global characterization of tropospheric $NO_2$ columns at specific times of the
day (Duncan et al., 2013; Veefkind et al., 2012) that have been applied to infer ground level $NO_2$
concentrations (Anenberg et al., 2022; Lamsal et al., 2011; Geddes and Martin, 2017; Cooper et
al., 2022). The emerging geostationary constellation (Al-Saadi et al., 2017) including the
Geostationary Environmental Monitoring Spectrophotometer (GEMS) over Asia, Tropospheric
Emissions: Monitoring Pollution (TEMPO) over North America, and Sentinel-4 over Europe
offers the prospect of inferring spatially comprehensive maps of hourly ground-level $NO_2$
concentrations. Towards this goal, there is need to develop an accurate representation of the hourly
$NO_2$ column to surface relationship.

Understanding the hourly variation of the relationship of $NO_2$ columns with surface

concentrations is of particular interest due to its role in the inference of hourly surface $NO_2$ from
satellite remote sensing. Numerous studies have separately examined the role of processes such as
surface emissions, boundary layer mixing, chemistry, deposition, and advection (Yang et al.,
2023b; Herman et al., 2009; Ghude et al., 2020; Zhang et al., 2016) upon the hourly variation of
$NO_2$ columns and upon surface $NO_2$ concentrations in the United States (Day et al., 2009), Spain



(Van Stratum et al., 2012), India (David and Nair, 2011), South Korea (Yang et al., 2023b, a) and
China (Tong et al., 2017). Differences have been identified in the daytime hourly variation of $NO_2$
tropospheric columns and surface concentrations during the DISCOVER-AQ and KORUS-AQ
campaigns with pronounced variation in surface concentrations that is not evident in the columns
(Choi et al., 2020; Crawford et al., 2021). Differences have also been noted in hourly variation of
$NO_2$ measured by aircraft and ground-based Pandora instruments (Li et al., 2021). There is a need
to understand the factors that can affect the relationship of hourly $NO_2$ columns with surface
concentrations.

Major challenges in the interpretation of satellite $NO_2$ observations include the short

lifetime of $NO_x$ (Laughner and Cohen, 2019), and localized emissions (Crippa et al., 2018) that
affect its spatial heterogeneity. Course resolution inputs to satellite retrieval algorithms (e.g.,
terrain height, albedo, and a priori $NO_2$ profiles) can lead to biases (Laughner et al., 2019;
Laughner et al., 2018; Russell et al., 2011). Complications with ground-based measurements of
the $NO_2$ columns as part of Pandora include uncertainties at steeper solar zenith angles during
morning and evening hours (Herman et al., 2009; Reed et al., 2015) and the changing Pandora
field of view (FOV) during morning and late evening (Li et al., 2021). Non-linearities in the
relationship between $NO_2$ concentrations and $NO_x$ sources or sinks can lead to biases in coarse-
resolution CTMs (Valin et al., 2011) that necessitate chemical transport models with a finer
resolution (Li et al., 2021, 2023a). Recent advances in the simulation of global atmospheric
composition at fine resolution (Eastham et al., 2018; Hu et al., 2018; Martin et al., 2022) offer the
opportunity to address the resolution need at the global scales of the geostationary constellation.

An important consideration in the inference of surface $NO_2$ concentrations with columnar



satellite observations is the vertical profile of $NO_2$ concentrations. Aircraft observations from the
NASA Deriving Information on Surface Conditions from Column and Vertically Resolved
Observations Relevant to Air Quality (DISCOVER-AQ) campaign offer measurements of the $NO_2$
vertical profile in the lower troposphere for evaluation of modeled vertical profiles (Flynn et al.,
2014; Reed et al., 2015). The Pandonia Global Network (PGN) is a global sun photometer network
that offers hourly measurements of total $NO_2$ columns (Verhoelst et al., 2021), useful for
interpretation of the daytime variation of $NO_2$ columns and evaluation of simulated columns. In
this study, we use the summertime $NO_2$ measurements from the NASA P-3B aircraft and Pandora
sun photometers over Maryland, Texas, and Colorado during the DISCOVER-AQ campaign to
understand the hourly variation of the $NO_2$ vertical distribution. We sample the high-performance
GEOS-Chem (GCHP) simulations along aircraft flight tracks and account for line-of-sight of the
Pandora sun photometers to interpret the hourly variation of $NO_2$ vertical distribution and vertical
columns. We explore the effect of hourly variation of temperature on the $NO_2$ cross-section, and
the Pandora columns. We further investigate the hourly variation of $NO_2$ columns and surface
concentrations from 50 PGN sites across the northern hemisphere. Section 2 describes the datasets
and methods used in this study to interpret the variation of $NO_2$ columns, surface concentrations,
and vertical distribution over DISCOVER-AQ and PGN sites. Section 3 examines the consistency
between the $NO_2$ vertical columns and surface concentrations across DISCOVER-AQ sites, and
PGN sites across the CONUS, Europe, and Asia. We explore the effects of model resolution and
boundary layer height adjustments on the hourly variation of $NO_2$ total columns and surface
concentrations as a function of hourly variation in mixed layer depth and photochemistry, and
measurement characteristics of Pandora sun photometers over PGN sites across the CONUS,
Europe, and Asia.



## 2      Materials and Methods

### 2.1      Aircraft measurements of $NO_2$ vertical profiles

The DISCOVER-AQ campaign offers comprehensive datasets of airborne and surface observations relevant for accessing air quality. One of the main objectives of the campaign was to examine the hourly variation of the relationship between the column and surface concentrations. In this study, we use aircraft, Pandora, and surface measurements over Maryland (July 2011), Texas (September 2013) and Colorado (July-August 2014) to investigate the hourly variation of $NO_2$ vertical profiles during summer when a long duration of daylight exists for analysis. Figure A1 shows the flight tracks, altitude variation, roadways and Pandora instrument locations over Maryland, Texas, and Colorado during the DISCOVER-AQ campaign. We focus on the aircraft spirals since they are designed to sample the vertical profile. We use $NO_2$ concentrations measured by the thermal dissociation laser-induced fluorescence (TD-LIF) technique (Thornton et al., 2000; Day et al., 2002) during the campaign. The laser-induced fluorescence method is highly sensitive for directly measuring $NO_2$, with a measurement uncertainty of 5 % and a detection limit of 30 pptv (Thornton et al., 2000). It also attempts to correct for positive interferences (Nault et al., 2015; Yang et al., 2023b). We use aircraft measurements from a height of about 300 m above ground level (AGL) up to 4 km AGL where high measurement frequency facilitates regional representation.

### 2.2      Pandonia Global Network $NO_2$ Total Column Densities

PGN is a global network of ground-based sun photometers that measures sun and sky radiance from 270 to 530 nm that allow retrievals of various trace gases including $NO_2$. Retrieval precision for total vertical $NO_2$ columns ("$NO_2$ columns" hereafter) is $5.4 \times 10^{14}$ molecules/cm$^2$ with a nominal accuracy of $2.7 \times 10^{15}$ molecules/cm$^2$ under clear-sky conditions (Herman et al.,



2009; Cede 2021). We use the level 2 data product from the version rnvs3p1-8 for PGN and for
DISOCOVER-AQ the data available from https://asdc.larc.nasa.gov/data/DISCOVER-AQ/. We
also include surface $NO_2$ observations from co-located DISCOVER-AQ and PGN sites. We use
$NO_2$ columns and surface concentrations employed during the DISCOVER-AQ campaign from
18 sites over Maryland, Texas and Colorado. We also include $NO_2$ columns and surface
concentrations from 50 PGN sites (the US: 31, Europe: 10, Asia: 9) for June-July-August (JJA)
2019. We focus on the $NO_2$ observations between 9 AM - 6 PM local solar time, for consistency
in observation frequency across all PGN sites. Tables A1 and A2 contain the names and location
of the DISCOVER-AQ and PGN sites respectively. We exclude Pandora measurements with
SZA>80°. We use total $NO_2$ columns including the stratosphere because the use of external
information sources to remove the stratospheric $NO_2$ columns from PGN can introduce errors in
the residual tropospheric columns (Choi et al., 2020).
**2.3    GEOS-Chem simulations**

We use GCHP, the high-performance configuration of the GEOS-Chem model that

operates with a distributed-memory framework for massive parallelization (Eastham et al., 2018),
to interpret the $NO_2$ column, vertical distribution and surface observations. GCHP offers the ability
to simulate the entire atmospheric column needed to interpret Pandora measurements at a fine
spatial resolution needed to interpret aircraft measurements. GEOS-Chem is driven by assimilated
meteorological data from the NASA Goddard Earth Observation System (GEOS). GEOS-Chem
includes a comprehensive $O_x$-$NO_x$-VOC-halogen-aerosol chemical mechanism in the troposphere,
in addition to the unified tropospheric-stratospheric chemistry extension in the stratosphere
(Eastham et al., 2014). We use GEOS-Chem 14.1.1 which includes recent updates to GCHP
(Martin et al., 2022), $NO_x$ heterogenous and cloud chemistry (Holmes et al., 2019), isoprene

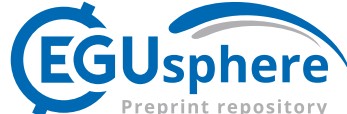

chemistry (Bates and Jacob, 2019), and aromatic chemistry (Bates et al., 2021). The ISORROPIA
II module simulates the thermodynamic partitioning between the gas and condensed phase
(Fountoukis and Nenes, 2007). Natural emissions include biogenic VOCs (Weng et al., 2020),
lightning $NO_x$ (Murray et al., 2012), and soil $NO_x$ (Weng et al., 2020). GEOS-Chem includes an
updated aircraft NOx emissions inventory for 2019, developed with the Aircraft Emissions
Inventory Code (Simone et al., 2013). Figure A2 shows the hourly variation of $NO_x$ emissions
across the PGN sites. For interpretation of PGN measurements in 2019, we conduct the simulations
for the year 2019 using GEOS-FP meteorology and the stretched grid capability (Bindle et al.,
2021) at a cubed sphere resolution of C180 (~55 km) and stretch factor of 4.0 yielding a regional
refinement of ~12 km. All simulations were conducted with a two-week spin-up. We interpolate
hourly GCHP outputs of simulated $NO_2$ columns and surface concentrations to the local solar time
at the PGN observation sites.

For interpretation of the DISCOVER-AQ aircraft campaigns, we conduct stretched grid

simulations over Maryland (July 2011), Texas (September 2013) and Colorado (July-August 2014)
with identical stretched grid configurations, with sampling along the aircraft flight tracks. We use
MERRA-2 meteorology for these simulations as GEOS-FP meteorology datasets are not available
prior to 2014. A sensitivity test for the year 2019 using either GEOS-FP and MERRA-2 affects
the local simulated $NO_2$ columns and surface concentrations by less than 5% for both 12 km and
55 km resolutions.

Hourly variation of the planetary boundary layer height (PBLH) can influence the vertical

distribution and hence the surface concentration of aerosols and trace gases (Lin and McElroy,
2010). Millet et al., (2015) found that GEOS-FP reanalysis over-estimates daytime PBLH as
compared to observations; correcting for PBLH estimations can lead to better agreement of ozone



(Oak et al., 2019) and PM$_{2.5}$ (Li et al., 2023b) with measurements. Our base case simulation uses
the PBLH derived from the Aircraft Meteorological Data Reports (AMDAR) at 54 sites across the
CONUS to adjust the PBLH estimates as described in Li et al., (2023). We examine the effect of
using the adjusted PBLH for simulations over the CONUS, Europe and East Asia. Table 1 shows
the 3 simulation cases conducted over Maryland, Texas, Colorado, the CONUS, Europe and East
Asia.
Table 1. Summary of GCHP Simulations

| Name | Description |
|---|---|
| Base_12 | 12 km base |
| NoΔBL_12 | 12 km without PBLH modification |
| NoΔBL_55 | 55 km without PBLH modification |

**2.5     Effective temperature of Pandora measurements**
The NO$_2$ cross section is temperature dependent with the magnitude of spectral features in a 294
K NO$_2$ spectrum about 80% of those in 220 K NO$_2$ spectrum (Vandaele et al., 2002). Thus, the
NO$_2$ columns fitted with a 220 K NO$_2$ spectrum are about 80% of those fitted with the 294 K NO$_2$
spectrum. Prior studies have identifies biases in the Pandora total ozone column effective
temperature driven by variation in seasonal temperature (Zhao et al., 2016; Herman et al., 2015).
We compare Pandora NO$_2$ effective temperatures with for the site-specific vertical variation of
hourly effective temperature using hourly GEOS-FP temperature profiles and GCHP NO$_2$ vertical
profiles  following equation (1) of Herman et al. (2009):
$$T_{eff} = \frac{\sum_i^N (\sigma(NO_2)_i \cdot VC(NO_2)_i \cdot (T)_i))}{\sum_i^N (\sigma(NO_2)_i \cdot VC(NO_2)_i))} \tag{1}$$

**2.6     Local solar time along Pandora line-of-sight**



At observing scenarios with large solar zenith angles, the Pandora sun photometer observes air
masses with varying local solar time at different altitudes along the line-of-sight. This feature is
particularly noteworthy for comparing hourly Pandora observations with other measurements or
simulations. Figure 1 shows the sampling process of GCHP simulations along the line-of-sight of
the Pandora sun photometer GCHP grid boxes are integrated along the viewing geometry of the
Pandora instrument to create a "staircase column" that accounts for effects of local solar time on
the horizontal and vertical variation in $NO_2$ concentrations. The variation in local solar time is
most relevant near sunrise and sunset when the $NO_2$/NOx ratios changes rapidly as discussed in
section 3.2. We correct the vertical columns reported by PGN to the local solar time of the
instrument by applying the ratio of integrated staircase columns to vertical columns.

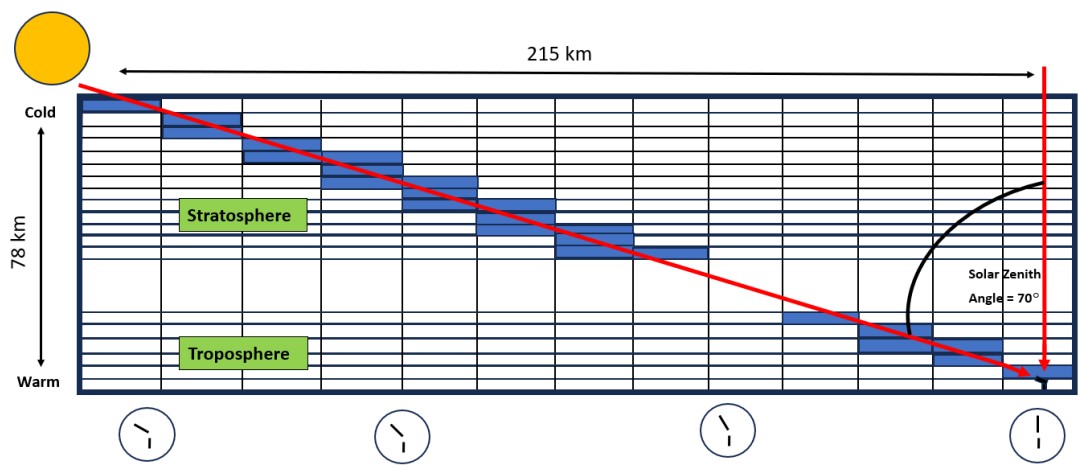


Figure 1. Configuration of integrating the GCHP grid boxes along the line-of-sight of the Pandora
instrument. The shaded grid boxes represent the line-of-sight for the Pandora sun photometer at an inclined
solar zenith angle. Clock faces indicate change in local solar time.
**2.7    Ground based surface $NO_2$ measurements**
We use hourly $NO_2$ surface concentrations from the catalytic converter measurements over
DISCOVER-AQ and PGN sites. Based on the approach of Lamsal et al., (2008) and Shah et al.,



(2020), we correct the interference of organic nitrates and $HNO_3$ in the $NO_2$ measurements, using
a correction factor derived from GCHP simulated site-specific $NO_2$, organic nitrates, and $HNO_3$
mixing ratios. The correction for $HNO_3$ and organic nitrates reduced the summertime mean $NO_2$
surface concentrations by 18% over DISCOVER-AQ sites and 23% over PGN sites.

## 3      Results and Discussion

### 3.1      Hourly variation of observed and simulated $NO_2$ vertical profiles

Figure 2 shows the hourly variation in the airborne TD-LIF measurements and simulated $NO_2$
vertical profiles at 12 km resolution (Base_12) over Maryland, Texas and Colorado during the
DISCOVER-AQ campaign. The measurements exhibit a pronounced maximum at 500 m at 10
AM (squares) that diminishes by a factor of 2 in the afternoon as concentrations become more

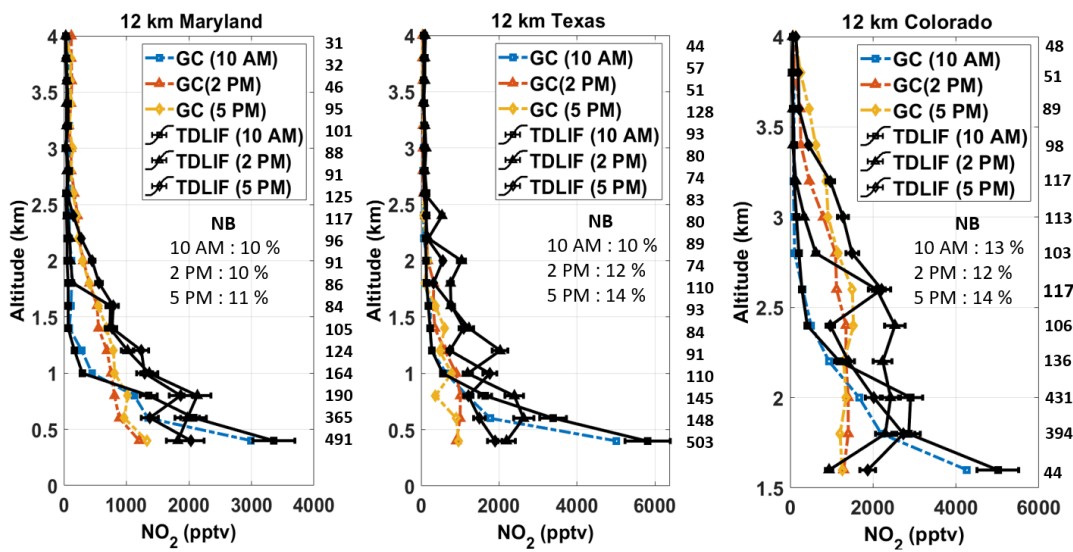


Figure 2: $NO_2$ vertical profiles from TD-LIF instrument aboard P-3B during the DISCOVER-AQ campaign over Maryland, Texas and Colorado. The black lines represent observations (square: 10 AM, triangle: 2 PM, diamond: 5 PM). The colored lines represent 12 km GCHP simulated mixing ratios (blue: 10 AM, orange: 2 PM, yellow: 5 PM). The inset values in the boxes show the normalized biases (NBs) at 10 AM, 2 PM, and 5 PM. The numbers on the right of each panel represent the number the observations associated with the corresponding altitude level. Error bars indicate standard errors in measurements.

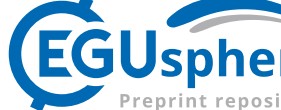



uniform below 1.5 km (triangles and diamonds), driven by the hourly variation in PBLH mixing
from early morning to late afternoon. For all three DISCOVER-AQ campaigns, the 12 km
simulated $NO_2$ mixing ratios (left panel) represent the vertical profile well with normalized bias
(NB) below 16% at local times: 10 AM, 2 PM, and 5 PM. The simulated $NO_2$ vertical profiles at
12 km without PBLH modifications (No$\Delta$BL_12) are similar (Figure A3). Figure A4 shows the
55 km simulated $NO_2$ vertical profiles (No$\Delta$BL_55). The 55 km GCHP simulations have increased
NB by a factor of 2, as compared to at 12 km. Overall, the $NO_2$ vertical profile exhibits greater
consistency with observations at 12 km than at 55 km by better resolving the heterogeneous
conditions along the aircraft flight tracks.
**3.2    Corrections to Pandora Effective Temperature**
The left panel in Figure 3 shows the Pandora and simulated mean hourly effective temperature of
the $NO_2$ columns over all PGN sites during June-August as inferred using hourly GEOS-FP

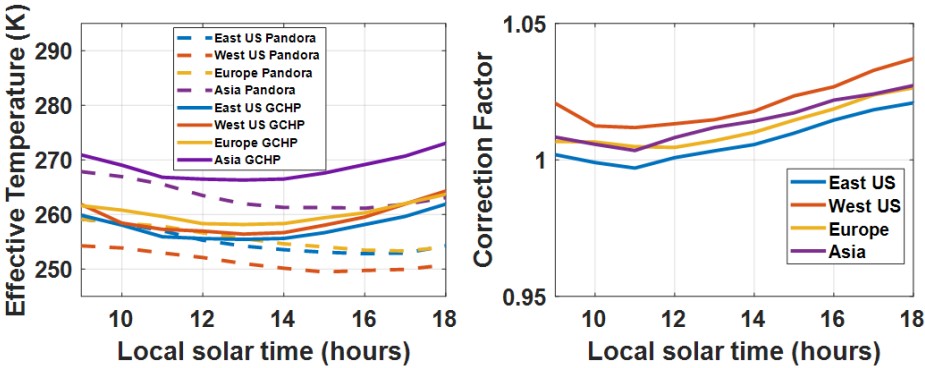


Figure 3. Hourly variation of the total $NO_2$ column mean effective temperature across all PGN sites (left
panel) and the corresponding correction factors (right panel).
temperature profiles and GCHP $NO_2$ vertical profiles. The simulated effective temperature is
lowest in early afternoon when near-surface $NO_2$ concentrations are minimum such that
stratospheric $NO_2$ makes a larger fractional contribution to the total column. The simulated



257 effective temperature deviates from the Pandora effective temperature with an increase toward

258 sunrise and sunset with increasing near-surface $NO_2$ fraction. The corresponding correction factor

259 (CF) for hourly variation in the effective temperature calculated as:

260     $$CF = 1 + \frac{0.2(T_{eff}(GCHP(hour)) - T_{eff}(Pandora(hour)))}{294 - 220}$$   (2)

261 The CF for the Pandora $NO_2$ columns increases toward sunrise and sunset due to the increased

262 effective temperature, reflecting the greater abundance of $NO_2$ molecules observed per unit

263 absorption. We apply site-specific CFs across all Pandora observations.

264 **3.3  Hourly variation of observed and simulated $NO_2$ VCDs**

265    Figure 4 (left) shows the mean hourly daytime Pandora vertical $NO_2$ columns summarized

266 from the summertime DISCOVER-AQ campaign measurements. The raw Pandora $NO_2$ columns

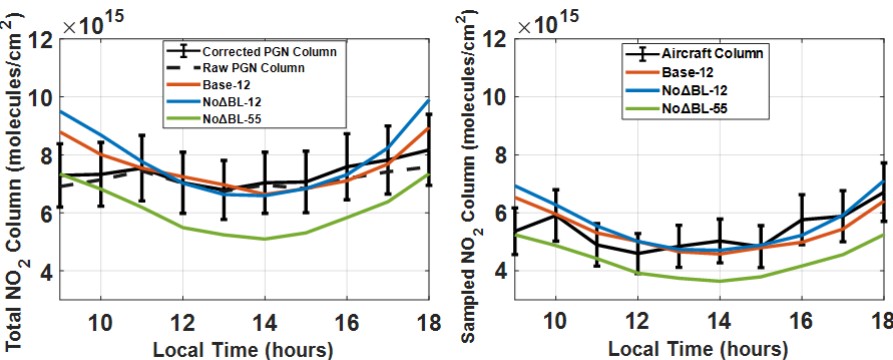

267
268
269 Figure 4. The left panel shows the total $NO_2$ vertical columns from corrected Pandora columns (black), raw
270 Pandora columns (black dotted), the 12 km base case simulation with staircase columns (red),12 km without
271 modified PBLH (blue) and 55 km without modified PBLH (green), during the DISCOVER-AQ campaigns
272 over Maryland (2011), Texas (2013) and Colorado (2014). The corrected PGN columns account for the
273 hourly variation in the effective temperature and the local solar time along the line-of-sight. The right panel
274 shows sampled aircraft and simulated partial columns (300 m A.G.L - 4 km A.G.L). Error bars indicate
275 standard error.
277 exhibit weak hourly variation of $8 \times 10^{14}$ molecules cm$^{-2}$ (within 10% of the daytime mean) that

278 is inconsistent with the aircraft measurements that indicate total columns in the morning and



evening of about $1.5 \times 10^{15}$ molecules cm$^{-2}$ greater than afternoon. The corrected Pandora
measurements that account for hourly variation in effective temperature and local solar time along
line-of-sight exhibit greater NO$_2$ columns in morning and evening by about $1.3 \times 10^{15}$ molecules
cm$^{-2}$, similar to the aircraft measurements. Since the Pandora instruments track the sun, viewing
stratospheric air masses 100 - 200 km away from the measurement station to the East in the
morning and to the West in the evening, the local solar time of stratospheric NO$_2$ observed by
Pandora instruments near sunrise and sunset is systematically shifted by about 5-10 mins towards
noon. This shift can be particularly important during sunrise and sunset when NO$_2$ columns in the
stratosphere undergo pronounced increase driven by an increasing NO$_2$/NOx ratio (Figure A5).
The 12 km simulated vertical columns generally represent the corrected Pandora observed columns
with an NB of 10%. Excluding the PBLH modification would have increased the NB to 13%.
Using a coarser 55 km simulation would have further degraded the agreement with an NB of 19%.
The hourly variation of partial NO$_2$ columns over 300 m to 4 km AGL from aircraft observations
exhibits a distinct increase in morning and evening and are well represented by the 12 km base
case simulation (NB =13%). Similar to our analysis for Pandora sites, excluding the PBLH
modification and coarsening the simulation to 55 km degrades the performance (NB = 15% and
25%) versus aircraft columns.
Figure 5 extends our analysis to all PGN sites across the CONUS, Europe and East Asia. Raw
measurements across all regions exhibit weak hourly variation. The correction for effective
temperature and local solar time along Pandora line-of-site increases the mean NO$_2$ columns in the
morning and evening by about $6 \times 10^{14}$ molecules cm$^{-2}$ across all regions. The base case simulation
generally reproduces measurements with NB of 9% for the eastern US, 14% for western US, 15%
for the Europe and 21% for east Asia sites. Excluding the PBLH correction would have increased





the NB (eastern US: 12%, western US: 18%, Europe: 18%, and eastern Asia: 26%) with the largest
change in Asia. Excluding the PBLH correction yields a higher daytime PBLH resulting in
increased chemical lifetime of $NO_x$, reduced $NO_2$ dry deposition rates and increased $NO_2/ NO_x$
ratio during afternoon and evening (Figure A6), thus leading to an hourly variation that deviates

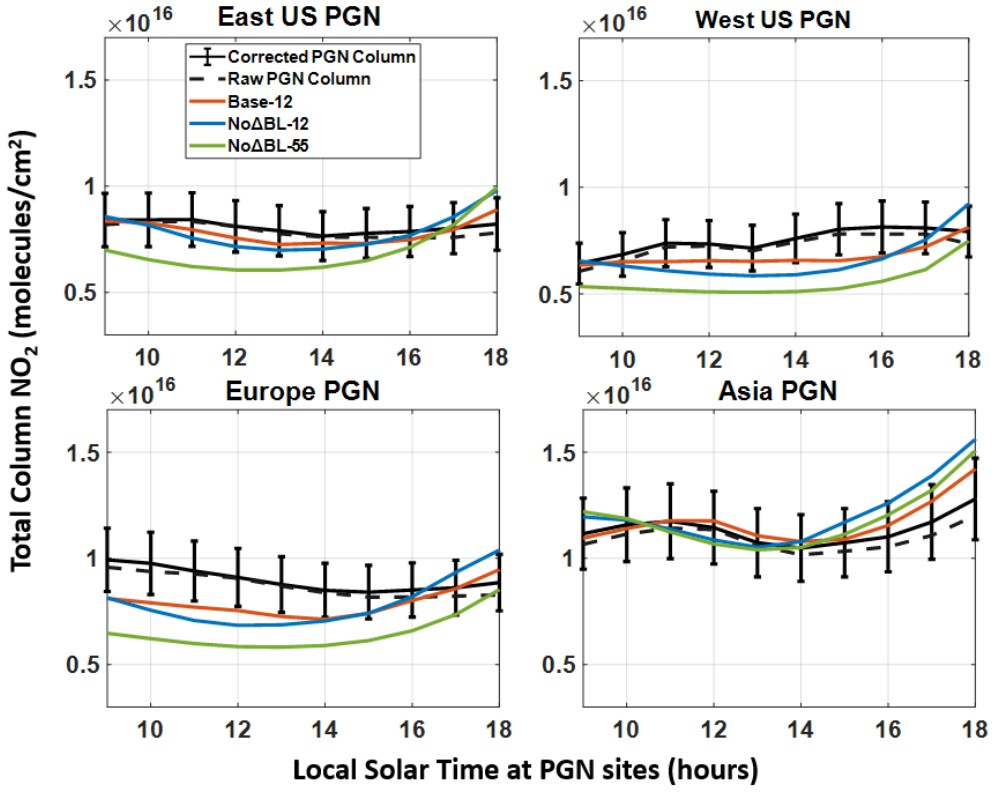


Figure 5. The total $NO_2$ vertical columns from corrected Pandora columns (black), raw Pandora columns
(black dotted), the 12 km base case simulation with staircase columns (red), 12 km without modified PBLH
(blue) and 55 km without modified PBLH (green) sampled over PGN sites for the summer months of June-
July-August in 2019. Error bars indicate standard error.
from the Pandora observations. Coarser resolution generally further increased the bias, reflecting
resolution effects discussed in the next section. The increase of the simulated total $NO_2$ columns
between 3-6 PM across all PGN sites reflects an increase in the $NO_2/NO_x$ ratio throughout the
column, driven by a reduction in $HO_x$ (Figure A7).



## 3.3 Simulated total NO₂ columns

Figure 6 shows the 12 km and 55 km simulated total $NO_2$ columns, for the summer months of

June-July-August in 2019, between 9 AM and 6 PM (local solar time) over the CONUS. The

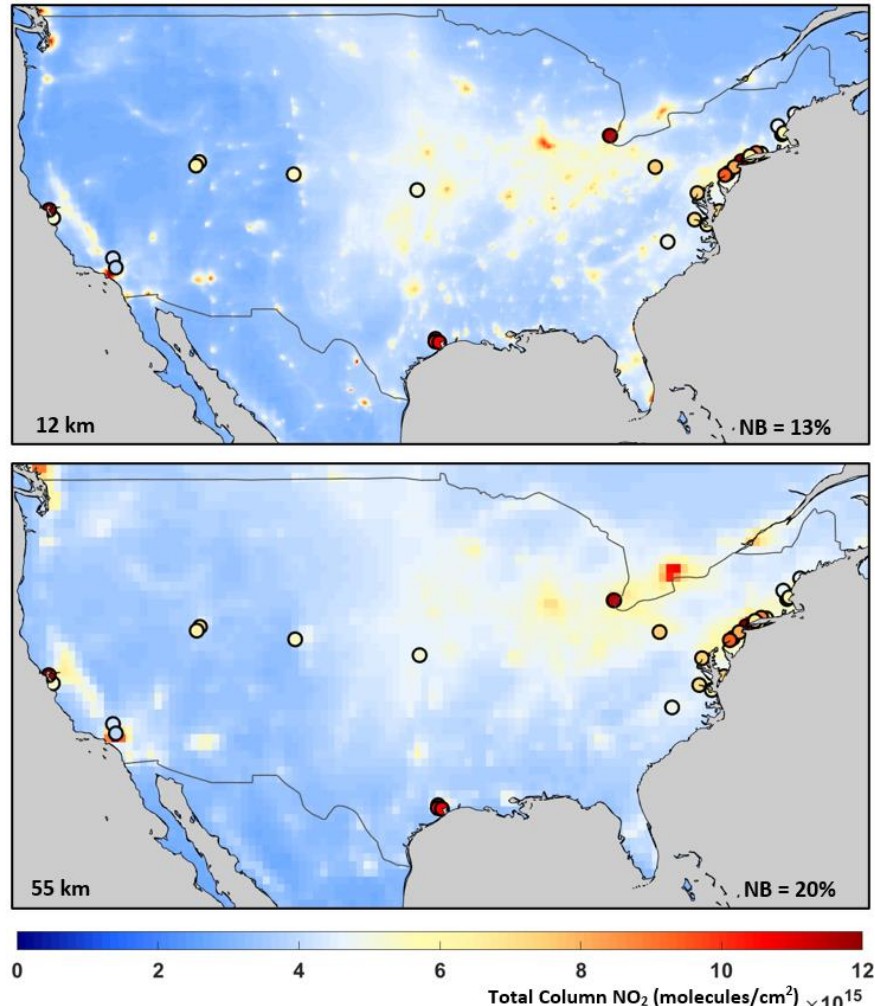

Figure 6. Simulated $NO_2$ total columns at 12 km (panel A) and 55 km (panel B) horizontal resolutions for the three-month average of June-July-August 2019 over domains where PGN monitors were available between 9 AM – 6 PM local solar time. The solid circles represent the PGN mean total columns between 9 AM – 6 PM local solar time for PGN sites in CONUS (31)

overlaid circles show the PGN mean total $NO_2$ columns. The 12 km simulated $NO_2$ columns



exhibit greater heterogeneity and better consistency with PGN observed columns (NB = 13%) as
compared to the 55 km simulated $NO_2$ columns (NB =20%). This is primarily driven by better
representation of emission and chemical processes at fine resolution (Zhang et al., 2023; Li et al.,
2023a ). Emissions at these sites are dominated by the transportation sector (Table A3). Figure 7
shows the total $NO_2$ columns from PGN, 12 km and 55 km for the summer months of June-July-

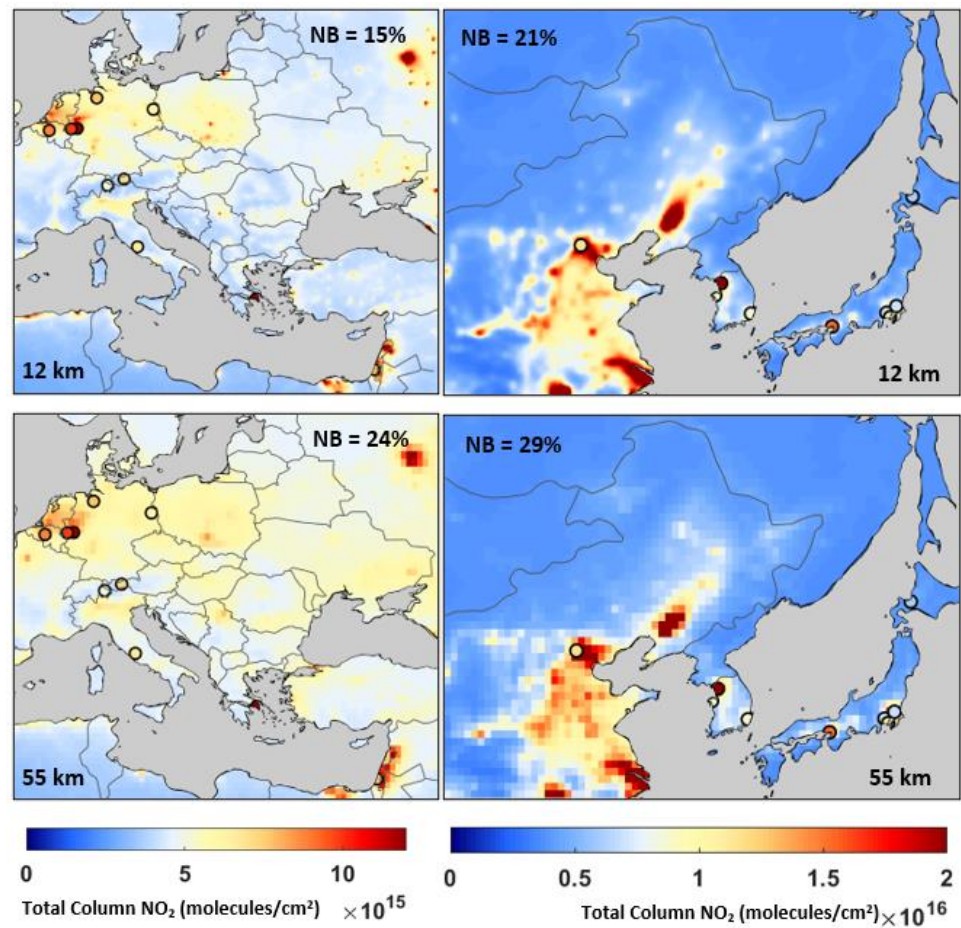


Figure 7. Simulated $NO_2$ total columns at 12 km (panel C and D) and 55 km (panel E and F) horizontal
resolutions  for the three-month average of June-July-August 2019 over domains where PGN monitors were
available between 9 AM – 6 PM local solar time.  The solid circles represent the PGN mean total columns
between 9 AM – 6 PM local solar time for the PGN sites in Europe (10) and Asia (9).

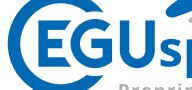

August in 2019, between 9 AM and 6 PM local solar time over Europe and East Asia. We find
enhanced $NO_2$ vertical columns over urban areas in western Europe, eastern China, Japan and the
Korean peninsula. The 12 km simulated $NO_2$ columns exhibit more resolved combustion features
and better agreement with Pandora observed columns for Europe (NB = 15%) and east Asia (NB
=17%) as compared to the 55 km simulated $NO_2$ columns for Europe (NB =24%) and east Asia
(NB =29%).
**3.4**      **Hourly variation of observed and simulated surface $NO_2$ concentrations**
Figure 8 shows the hourly variation in surface $NO_2$ mixing ratios from the corrected in situ
measurements and 12 km simulations over Maryland, Texas and Colorado. Measured $NO_2$ mixing
ratios are greater in morning and evening than in afternoon as expected from the mixed layer
growth and shorter $NO_x$ lifetime in afternoon. Observed $NO_2$ surface concentrations over PGN
sites in Asia show enhancement at evening hours (5-6 PM) as compared to PGN sites elsewhere.

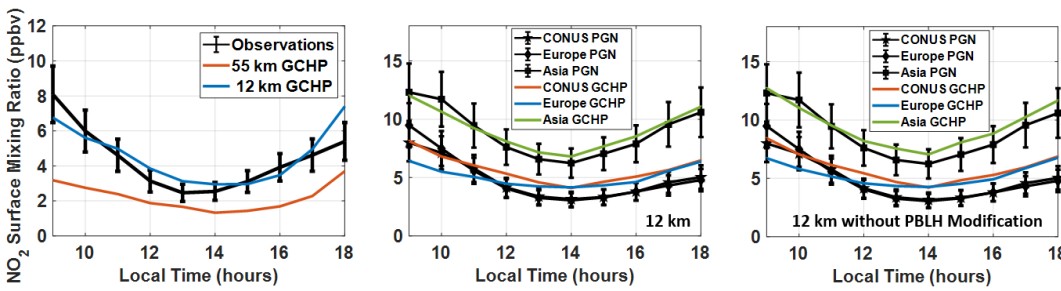


Figure 8. The left panel shows the hourly variation of corrected surface $NO_2$ mixing ratios from
observations during the DISCOVER-AQ campaign. The middle and right panels show the hourly variation
of observed and 12 Km PBLH modified and 12 km simulated surface $NO_2$ mixing ratios averaged over the
PGN sites respectively. Error bars indicate standard error.
The measurements are better represented at 12 km (NB = 21%) than at 55 km (NB =63%) by better
resolving high $NO_x$ emissions near measurement sites. Both Base_12 and No$\Delta$BL_12 simulated
$NO_2$ concentrations generally represent the observations well with NB = 18% (Base_12) and NB



= 20% (NoΔBL_12), across all PGN sites.

## 3.5    Hourly variation of layer contributions to total $NO_2$ staircase columns

We proceed to apply the 12 km simulations to understand how the $NO_2$ vertical profile affects the
$NO_2$ column to surface relationship. Figure 9 shows the hourly variation of simulated contributions
to the $NO_2$ total columns (Base_12) from different vertical layers for multiple regions. In all four
regions, within the troposphere, the layer below 0.5 km is the largest contributor at 9 AM with a
diminishing contribution into the afternoon associated with mixed layer growth and an increasing
contribution towards evening. The contribution from layers between 0.5 km and the tropopause

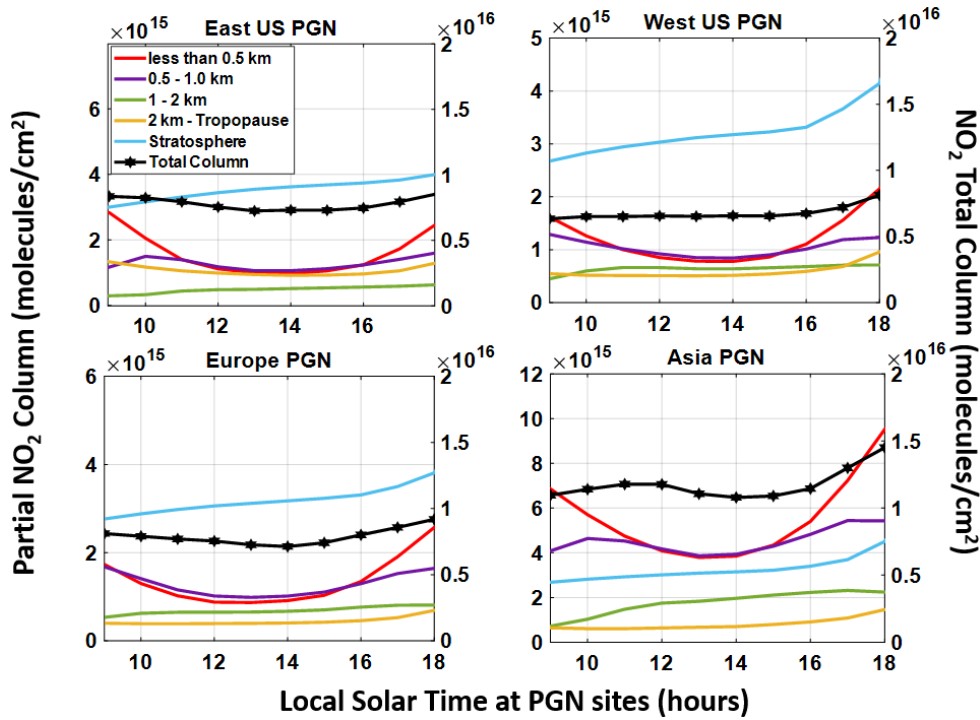

Figure 9. The simulated absolute contribution of $NO_2$ columns at different hours of the day averaged over
the summer months of June-July-August for 2019 for PGN sites over the eastern US, western US, Europe,
and eastern Asia. The colored lines resemble the absolute concentrations from different sections of the
column. The black line (hexagon) represents the total $NO_2$ column. The right y-axis (specifically for the
total $NO_2$ column representing the black marked line) shows the total columns of $NO_2$.



has weaker variation contributing to the overall weaker variation in total columns. Fractional layer
contributions are shown in Figure A8. Fractional hourly variation of the layers above 0.5 km
exhibits a compensating inverse behavior, with a pronounced variation in the stratospheric
fraction. Contributions from the free troposphere are relatively high for the eastern US reflecting
the lightning contribution (Shah et al., 2023; Dang et al., 2023). Over Asia the fractional
contribution below 0.5 km is the highest (26% - 42%) reflecting major surface contributions.
Overall, we find that for all four regions, the hourly variation in the total column reflects hourly
variation below 500 m, dampened by greater column contributions above 500 m that dominate the
total column.
**Conclusion**
We applied the GCHP model to investigate the hourly variation of summertime $NO_2$ columns and
surface concentrations by interpreting DISCOVER-AQ aircraft and ground-based measurements
over Maryland, Texas, Colorado and PGN measurements over the CONUS, Europe, and eastern
Asia. We corrected the hourly variation in Pandora observations for the effects of temperature on
the $NO_2$ cross section and the local solar time along the Pandora line-of-sight. The site-specific
effective temperature correction factors typically increase the hourly variation of the Pandora
observed columns over DISCOVER_AQ sites (3.5% from the daytime mean) and PGN sites (4%
from the daytime mean). Near sunrise and sunset, differences in local solar time observed by
Pandora in the stratosphere versus the measurement site reflect displacement of 5-10 mins in local
solar time toward noon which is relevant in the stratosphere near sunrise and sunset when the
$NO_2/NO_x$ ratio is varying rapidly. These corrections to the Pandora measurements improve their
consistency with the hourly variation in the $NO_2$ columns inferred from DISCOVER-AQ aircraft
measurements. We find that the fine scale simulations at 12 km better represent the $NO_2$ vertical



profile measured by aircraft, reducing the NB from 23% to 16% as compared to the simulations at
moderate resolution of 55 km. Simulations at fine resolution (~12 km) of vertical columns along
the line-of-sight of Pandora instruments agree better with Pandora sun photometers at
DISCOVER-AQ sites (10%), and across the eastern US (9%), western US (14%), Europe (15%)
and Asia (21%) as compared to moderate resolution (55 km). Fine resolution represents
atmospheric physical and chemical processes with greater accuracy. Excluding the effects of
model resolution, and the PBLH modification increases the NB to 21% across DISCOVER-AQ
sites (over Maryland, Texas and Colorado) and increases the NB at PGN sites over the eastern US
(17%), western US (24%) , Europe (24%) and east Asia (29%). Adjusting the PBLH to represent
observations improved the daytime variation in $NO_2/NO_x$ ratios by increasing the $NO_2/NO_x$ ratio
in midday and decreasing the $NO_2/NO_x$ ratio in afternoon and evening.
We use the simulated columns to derive the hourly contribution of vertical layers to the
total tropospheric columns. We find weaker hourly variation in $NO_2$ columns than in the lowest
500 m where $NO_2$ concentrations are greater in morning and evening than midday, while the
residual tropospheric column above 500 m dominates the total column with weaker variability.
Thus, the weak hourly variation in the column reflects fractional contributions from $NO_2$ below
and above 500 m. Future work should leverage the information developed here to test the
performance of surface $NO_2$ concentrations inferred from the geostationary constellation versus
ground-based measurements.

**Code and Data Availability**

GEOS-Chem 14.1.1 along with GCHP code is available for download at
https://github.com/geoschem/GCHP.git. The PGN data is available at https://data.pandonia-
global-network.org/. The DISCOVER-AQ aircraft and Pandora data are available here:



https://asdc.larc.nasa.gov/project/DISCOVER-AQ.   For hourly simulated NO₂ datasets please
contact the author (deepangsuchatterjee@wustl.edu; deepangsuchatterjee@gmail.com)

**Author contributions**

The manuscript was written using contributions from all authors. The conceptualization was
initialized by DC and RVM. The methodology was developed by DC and RVM .DC conducted
the model simulations. DC conducted the data analysis with help from CL,DZ,HZ,LL,DH,RC. JC
conducted the DISCOVER-AQ campaign. AC manages the PGN datasets. DC and RVM wrote
the original draft. All authors have reviewed, edited and given approval to the final version of the
manuscript.

**Competing interests**

The contact author has declared that neither they nor their co-authors have any competing interests.

**Acknowledgments**

This work has been supported by the NASA Grant 80NSSC21K1343 and 80NSSC21K0508 and
NSF Grant 2244984. We thank the GEOS-Chem support team for maintaining the model used in
this work.

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

664                                        **Appendix**

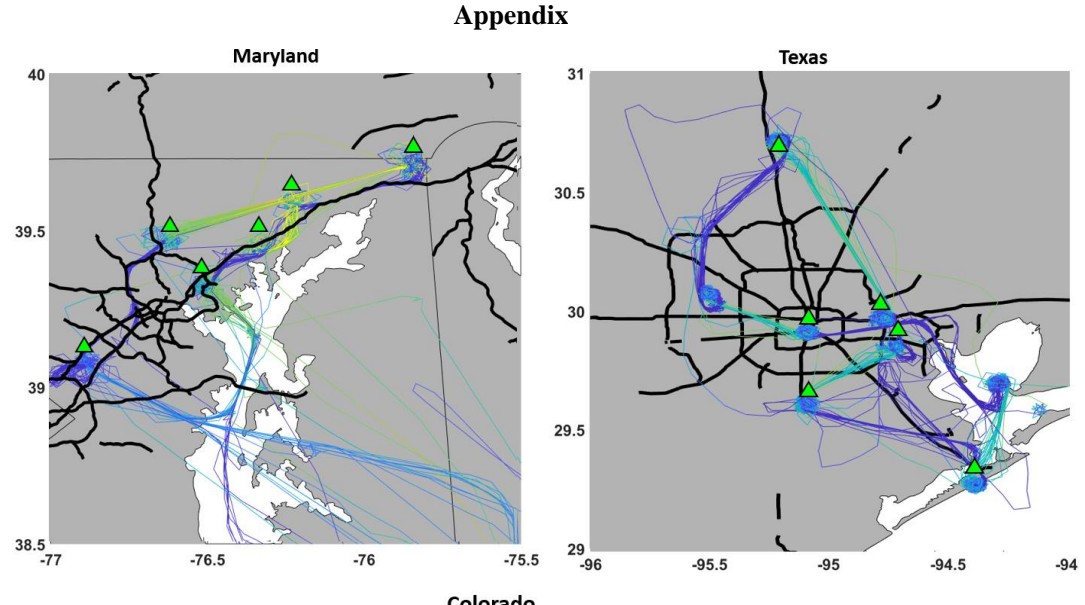


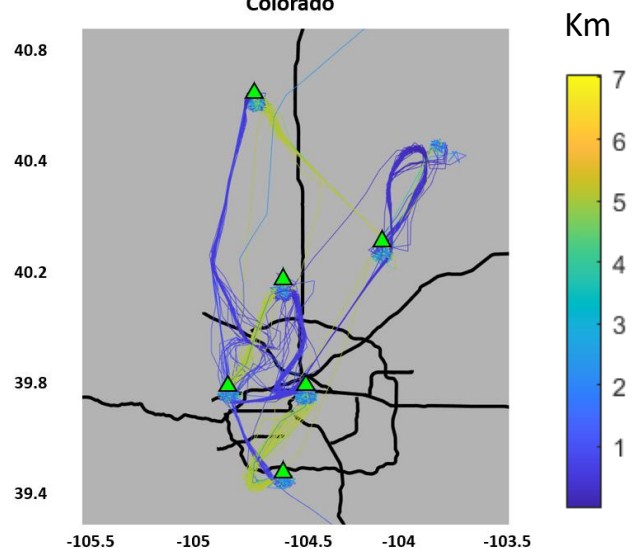





Figure A1. Flight tracks showing the path and altitude of the P-3B aircraft during the DISCOVER-AQ campaign over Maryland during July 2011 (left), over Texas during September 2013 (center) and Colorado (right). The green triangles show the locations of the Pandora sun photometers that have been used in this study. The Sites names and coordinates are listed in Table A1. Grey indicates land, white indicates water. The black bold lines indicate roads.

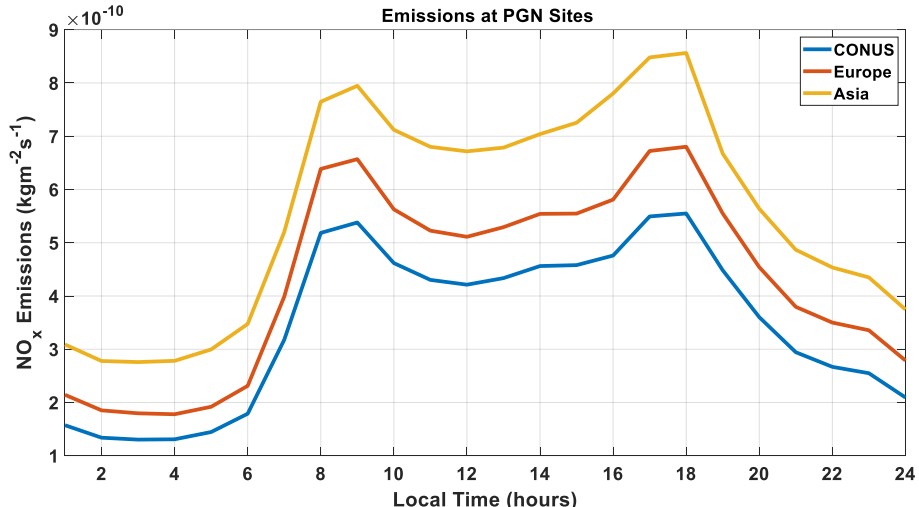

Figure A2. Hourly variation of NOx emissions including all sectors across 50 PGN sites over the CONUS, Europe, and east Asia.

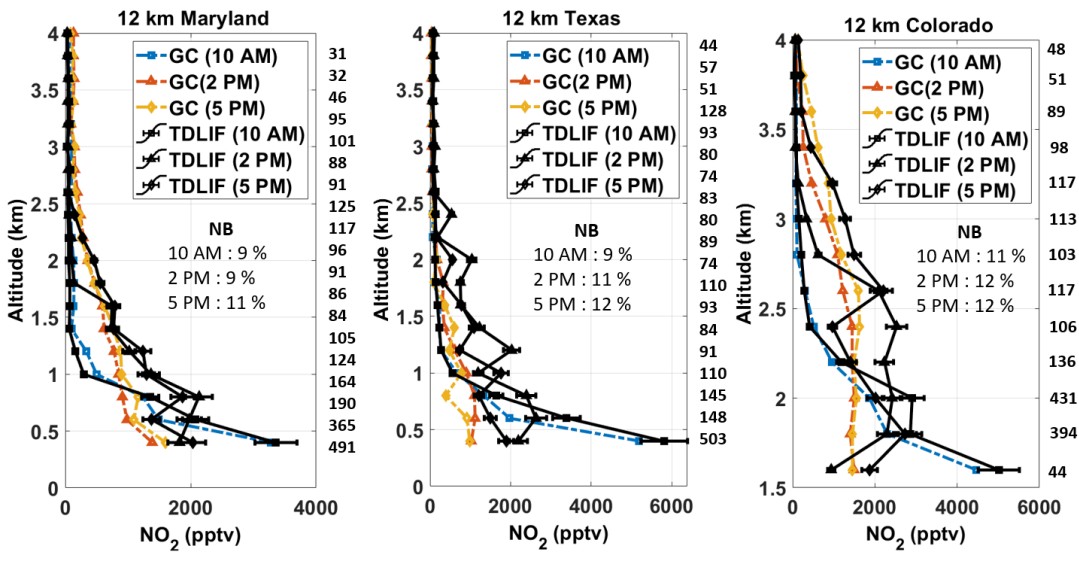



Figure A3: NO₂ Vertical profiles from TD-LIF instrument aboard during the DISCOVER-AQ campaign
over Maryland, Texas and Colorado. The black lines represent observations (square: 10 AM, triangle: 2
PM, diamond: 5 PM). The colored lines represent GCHP 12 km simulated NO₂ mixing ratios without
modifying the PBLH  (blue: 10 AM, orange: 2 PM, yellow: 5 PM). The inset values in the boxes show the
NB at 10 AM, 2 PM, and 5 PM. The numbers in the middle represent the number the observations associated
with the corresponding altitude level. Error bars indicate standard errors in measurements.

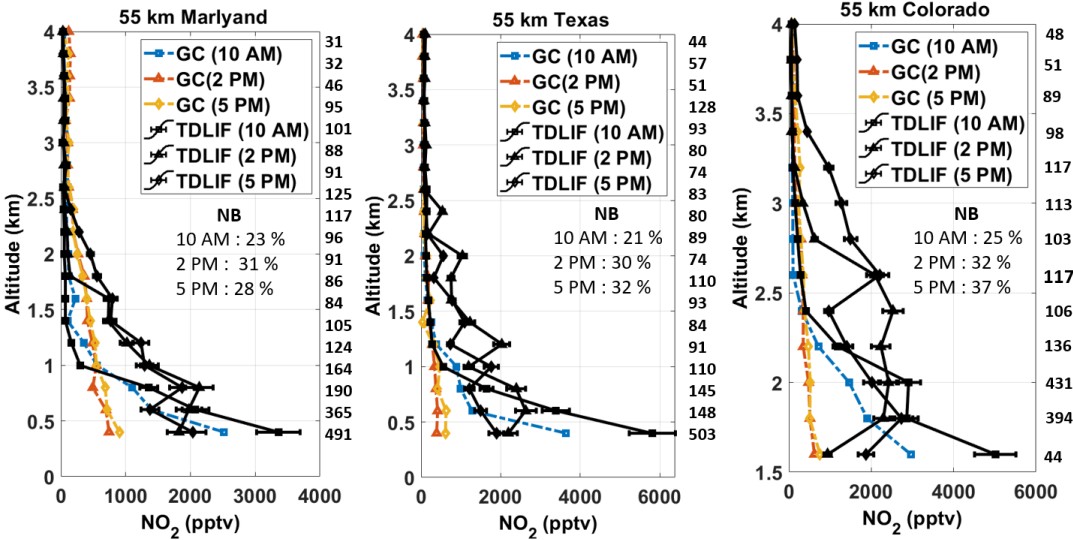


Figure A4: NO₂ Vertical profiles from TD-LIF instrument aboard during the DISCOVER-AQ campaign
over Maryland, Texas and Colorado. The black lines represent observations (square: 10 AM, triangle: 2
PM, diamond: 5 PM). The colored lines represent GCHP 55 km simulated NO₂ mixing ratios without
modifying the PBLH  (blue: 10 AM, orange: 2 PM, yellow: 5 PM). The inset values in the boxes show the
NB at 10 AM, 2 PM, and 5 PM. The numbers in the middle represent the number the observations associated
with the corresponding altitude level. Error bars indicate standard errors in measurements.



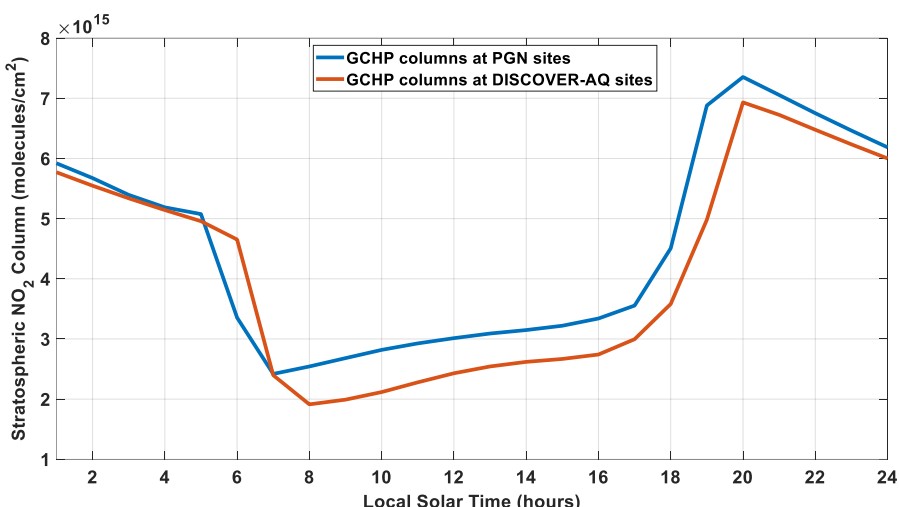


Figure A5. GCHP NO$_2$ stratospheric columns for the three-month average of June-July-August at DISCOVER-AQ sites (red) and PGN sites (blue).

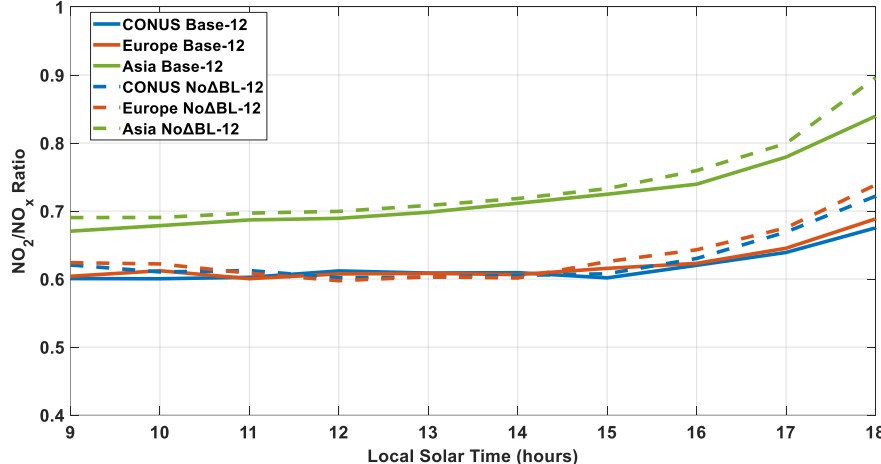


Figure A6. Hourly variation of 12 km simulated column NO$_2$/NOx ratios across 50 PGN sites over the CONUS (red), Europe (blue), and east Asia (green). The dotted lines show the 12 km simulated NO$_2$/NOx ratios without modified PBLH.



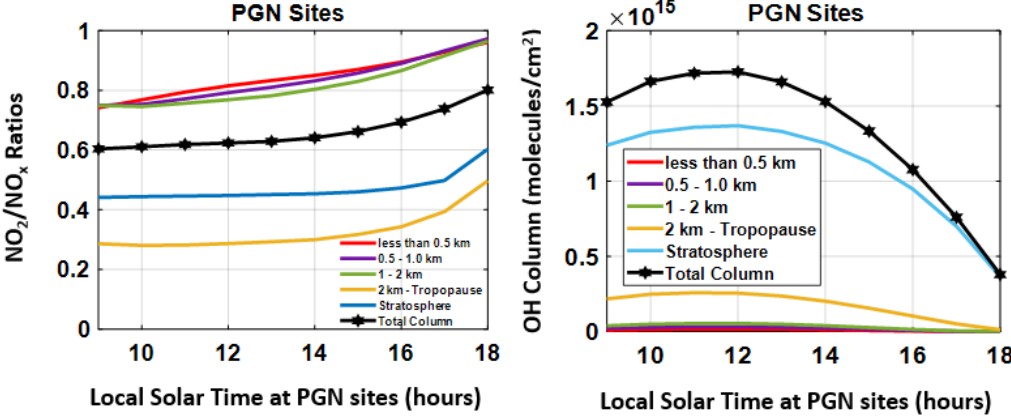


Figure A7. Simulated $NO_2/NO_x$ ratios (left panel) and simulated partial and total OH columns (right panel)
at different hours of the day averaged over the summer months of June-July-August for 2019 for PGN sites
over the eastern US, western US, Europe, and eastern Asia.


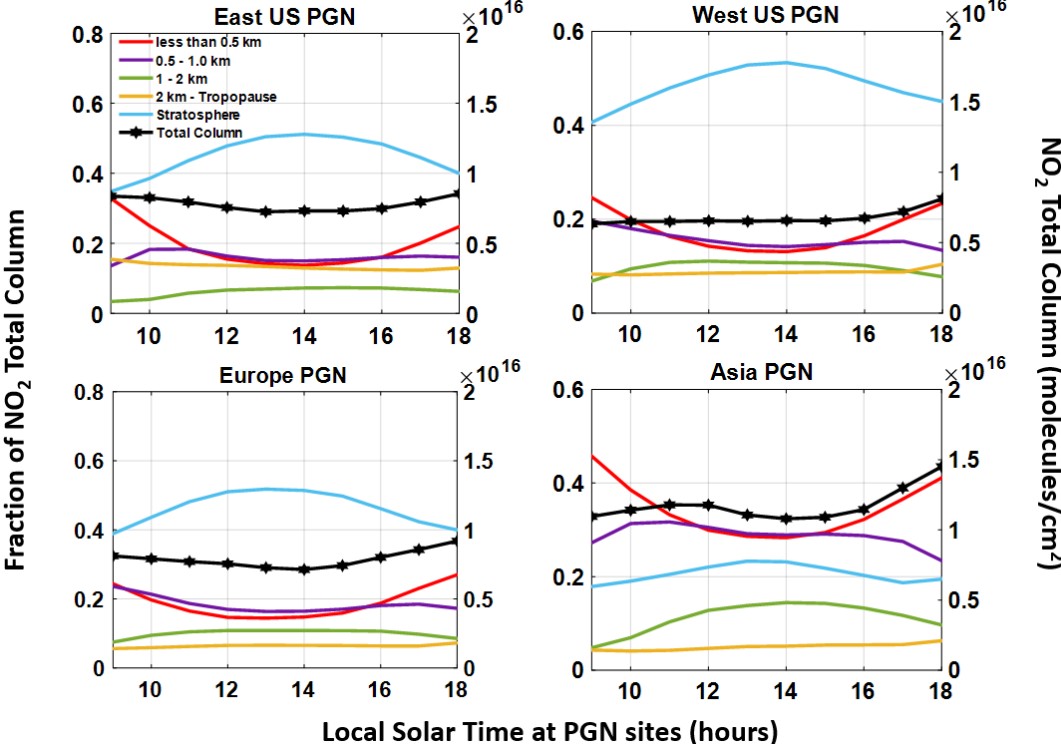


Figure A8. The simulated fractional contribution of $NO_2$ columns at different hours of the day averaged
over the summer months of June-July-August for 2019 for PGN sites over the eastern US, western US,
Europe, and eastern Asia. The right Y-axis shows the total columns of $NO_2$.






Table A1. Site name, latitude and longitude for 18 sites in Texas, Maryland, and Colorado that has
concurrent pandora and aircraft measurements.

| Site | Sites name | Latitude | Longitude | Date |
|------|-----------|----------|-----------|------|
| | **Texas Sites** | | | September 2013 |
| 1. | Channelview | 29.802 | 95.125 | |
| 2. | Conroe | 30.350 | 95.425 | |
| 3. | Deer Park | 29.670 | 95.128 | |
| 4. | Galveston | 29.254 | 95.861 | |
| 5. | Manvel Croix | 29.520 | 95.392 | |
| 6. | Moody Tower | 29.718 | 95.341 | |
| | **Maryland Sites** | | | July 2011 |
| 1. | Aldino | 39.563 | 76.204 | |
| 2. | Beltsville | 39.055 | 76.878 | |
| 3. | Edgewood | 39.410 | 76.297 | |
| 4. | Essex | 39.311 | 76.474 | |
| 5. | Fairhill | 39.701 | 75.860 | |
| 6. | Padonia | 39.461 | 76.631 | |
| | **Colorado Sites** | | | July-August 2014 |
| 1. | Bao Tower | 40.043 | 105.012 | |
| 2. | Chatfield Park | 39.535 | 105.074 | |
| 3. | Denver La Casa | 39.782 | 105.018 | |
| 4. | Fort Collins | 40.595 | 105.143 | |
| 5. | Platteville | 40.183 | 104.734 | |
| 6. | NREL-Golden | 39.743 | 105.181 | |


Table A2. Site name, latitude and longitude for 31 sites in CONUS and 11 sites in Europe, North

Africa and Middle-east, and 9 sites in east Asia from the PGN database.




| Site | Site Name | Latitude | Longitude | Date |
|---|---|---|---|---|
| | **Eastern US** | | | June-July-August 2019 |
| 1. | 'Bristol_PA' | 40.1074 | -74.8824 | |
| 2. | 'Cambridge_MA' | 42.3800 | -71.1100 | |
| 3. | 'CapeElizabeth_ME' | 43.5610 | -70.2073 | |
| 4. | 'ChapelHill_NC' | 35.9708 | -79.0933 | |
| 5. | 'CharlesCity_VA' | 37.3260 | -77.2057 | |
| 6. | 'Dearborn_MI' | 42.3067 | -83.1488 | |
| 7. | 'Detroit_MI.txt' | 42.3026 | -83.1068 | |
| 8. | 'Hampton_VA' | 37.0203 | -76.3366 | |
| 9. | 'Londonderry_NH' | 42.8625 | -71.3801 | |
| 10. | 'Lynn_MA' | 42.4746 | -70.9708 | |
| 11. | 'Madison_CT' | 41.2568 | -72.5533 | |
| 12. | 'Manhattan_NY' | 40.8153 | -73.9505 | |
| 13. | 'NewBrunswick_NJ' | 40.4622 | -74.4294 | |
| 14. | 'NewHaven_CT' | 41.3014 | -72.9029 | |
| 15. | 'OldField_NY' | 40.9635 | -73.1402 | |
| 16. | 'Philadelphia_PA' | 39.9919 | -75.0811 | |
| 17. | 'Pittsburgh_PA ' | 40.4655 | -79.9608 | |
| 18. | 'WallopsIsland_VA ' | 37.8439 | -75.4775 | |
| 19. | 'WashingtonDC' | 38.9218 | -77.0124 | |
| 20. | 'Westport_CT' | 41.1183 | -73.3367 | |



|  |  | **Western US** |  | June-July-August 2019 |
|---|---|---|---|---|
| 21. | 'Aldine_TX' | 29.9011 | -95.3262 | |
| 22. | 'Boulder_CO' | 40.0375 | -105.2420 | |
| 23. | 'Edwards_CA ' | 34.9600 | -117.8811 | |
| 24. | 'Houston_TX' | 29.7200 | -95.3400 | |
| 25. | 'LaPorte_TX' | 29.6721 | -95.0647 | |
| 26. | 'Manhattan_KS' | 39.1022 | -96.6096 | |
| 27. | 'MountainView_CA' | 37.4200 | -122.05680 | |
| 28. | 'Richmond_CA' | 37.9130 | -122.3360 | |
| 29. | 'SaltLakeCity_UT' | 40.7663 | -111.8478 | |
| 30. | 'SouthJordan_UT' | 40.5480 | -112.0700 | |
| 31. | 'Wrightwood_CA' | 34.3819 | -117.6813 | |
|  |  | **Europe** |  | June-July-August 2019 |
| 32. | 'Athens' | 37.9878 | 23.7750 | |
| 33. | 'Bremen' | 53.0813 | 8.8126 | |
| 34. | 'Brussels' | 50.7980 | 4.3580 | |
| 35. | 'Cologne' | 50.9389 | 6.9787 | |
| 36. | 'Davos' | 46.8000 | 9.8300 | |
| 37. | 'Innsbruck' | 47.2643 | 11.3852 | |
| 38. | 'Juelich' | 50.9080 | 6.4130 | |
| 39. | 'Lindenberg' | 52.2900 | 14.1200 | |
| 40. | 'Rome' | 42.1057 | 12.6402 | |
| 41. | 'Tel-Aviv' | 32.1129 | 34.8062 | |





| | | **Eastern Asia** | | June-July-August 2019 |
|---|---|---|---|---|
| 42. | 'Beijing' | 40.0048 | 116.3786 | |
| 43. | 'Kobe' | 34.7190 | 135.2900 | |
| 44. | 'Sapporo' | 43.0727 | 141.3459 | |
| 45. | 'Seosan' | 36.7769 | 126.4938 | |
| 46. | 'Seoul' | 37.5644 | 126.9340 | |
| 47. | 'Tokyo' | 35.6200 | 139.3834 | |
| 48. | 'Tsukuba' | 36.0661 | 140.1244 | |
| 49. | 'Ulsan' | 35.5745 | 129.1896 | |
| 50. | 'Yokosuka' | 35.3207 | 139.6508 | |


Table A3. Sectoral contribution to NOx emissions averaged over all PGN sites, the US, Europe
and Asia.

| PGN Sites | TRA(%) | RCO(%) | IND(%) | ENE(%) | SHP(%) | AGR(%) | WST(%) |
|---|---|---|---|---|---|---|---|
| ALL | 49 | 19 | 13 | 7 | 7 | 4 | 1 |
| CONUS | 45 | 29 | 16 | 4 | 2 | 3 | 1 |
| Europe | 47 | 11 | 8 | 10 | 16 | 7 | 1 |
| Asia | 55 | 12 | 15 | 9 | 4 | 3 | 2 |


TRA: Transport, RCO: Residential Combustion, IND: Industry, ENE: Energy, SHP: Ship Emissions, AGR:
Agriculture, WST: Waste