# Peer review of "Interpreting Summertime Hourly Variation of NO2 Columns with Implications for"

_EGUsphere, 2024_

## Referee Comment (RC2)

**General comment**

This study explores the diurnal variation of total $NO_2$ columns and surface $NO_2$ concentrations using GCHP model simulations with independent $NO_2$ measurements from DISCOVER-AQ campaign and Pandonia Global Network. Two corrections are applied on PGN/Pandora total $NO_2$ columns to better represent the dependence of $NO_2$ cross section on the temperature, and different local solar time along the PGN/Pandora line-of-sight. Besides, the authors also test the influences of horizontal resolution and planetary boundary layer height (PBLH) modification on the model performance against aircraft and ground-based observations. It is demonstrated that compared with the other two sensitivity runs, fine scale (~12km) modelled $NO_2$ columns with PBLH modification show smaller bias to independent measurements and better agreement in terms of $NO_2$ diurnal variation. Based on model simulations, the authors find that $NO_2$ columns below 500m show much stronger diurnal variation that that of total columns, which is dampened by residual columns above with much weaker variability.

The findings of this study are important for understanding the relationship between $NO_2$ columns and surface concentrations, and I recommend it to be published after addressing following issues.

**Specific comments**

Line 39: what is "connected layers"?

Line 200-203: please re-write this sentence and explain the meaning of each term in this equation.

Line 241: what do you mean by "left panel" and in which figure?

Line 254-255: why the simulated effective temperature is lowest in the early afternoon?

Line 258-260: please explain the scientific meaning of "0.2" and "(294-220)" in the equation.

Line 289: it seems that PBLH modification has larger impact on simulated $NO_2$ columns in the morning and evening than midday. What is the reason for this?

Line 292: in Figure 4 and 5, both corrected PGN/Pandora $NO_2$ columns and aircraft partial $NO_2$ columns present a distinct increase in early morning, which is also found in GEMS $NO_2$ observations. However, this feature is not reproduced in modelled $NO_2$ columns even though $NO_x$ emissions have a morning peak around 9:00 a.m. local time. What is the explanation for this discrepancy?

**Technical comments**

Line 31-35: please simplify this sentence.

Line 36: change "column" to "columns".

Line 70: please expand the abbreviation "KORUS-AQ" when it appears for the first time.

Line 85: please expand the abbreviation "CTMs" when it appears for the first time.

Line 97-102: please combine these two sentences to make it less redundant.

Line 108: please expand the abbreviation "CONUS" when it appears for the first time.

Line 136-137: please re-write this sentence.

Line 143: please keep consistent expression of longitude (sign) in Tables A1 and A2.

Line 198: change "identifies" to "identified".

Line 315: change "3.3" to "3.4".

Line 340: change "3.4" to "3.5".

Line 355: change "3.5" to "3.6".

Line 397: remove the comma.

---

## Author Comment (AC1)

The questions and comments from reviewers are in black, the responses are in black italics and the lines included in the main text are in blue.

**Reviewer 1**

The study investigates the diurnal variation of the NO2 column-to-surface-concentration relationships using observations and corresponding model simulations from three DISCOVER-AQ campaigns and worldwide Pandora NO2 total column measurements. The authors correct the Pandora measurements by considering biases in effective temperature and local solar timing along the line-of-sight used in Pandora retrieval. Regarding model simulations, the manuscript compares two simulations with different horizontal resolutions (12 vs. 55 km) to evaluate the impact of horizontal resolution on model performance in reproducing observed NO2 surface concentrations and columns. The authors demonstrate that NO2 above 500 m has weak temporal variability but dominates total NO2 columns, dampening the variation of total NO2 columns, although NO2 below 500 m shows apparent diurnal variation. The results are interesting and useful to the community.

*Response: We thank the reviewer for acknowledging the importance of this work and its importance to the community. Specific responses are listed below:*

However, I have two major concerns about the quality of the manuscript.

Major comments:

1. Section 3.5 is entirely based on model results. However, Figures 4 and 5 show that the model simulation results differ from the Pandora measurements in the diurnal variations of NO2 total columns even if the Pandora measurements have been slightly corrected. Are you assuming the model is correct? If so, can you explain why Pandora shows weaker diurnal variations than model results? According to the model results, even if NO2 below 500 m contributes less than 50% to total NO2 columns, we should still observe apparent diurnal variations in NO2 columns (Figures 4 and 5). Why doesn't it occur in Pandora measurements? On the other hand, if you assume Pandora is correct, how can Section 3.5 convince the community, considering that it is based on model results different from observations?

*Ans: We consider the improved Pandora observations to be correct within their uncertainty. As shown in Figures 4 and 5, the 12 km simulation is within the uncertainty of the Pandora measurements for nearly all times of day across all regions. Thus, we believe the simulation has sufficient skill for its application to interpret the total column behavior presented in Section 3.6 (previously 3.5). Despite this skill in the 12 km simulation, we agree with the reviewer that the simulation generally exhibits greater hourly variation than observed with Pandora. Part of this difference may reflect profile differences within 1-2 km above ground level as identified by the aircraft observations. We add to lines 242-244:*

"Differences tend to be larger within 1-2 km above ground level in afternoon (2 PM and 5 PM local time), which integrates to a lower simulated partial column of 6 x $10^{14}$ molecules cm$^{-2}$."

*Further reconciling the remaining differences remains elusive in part because the base case 12 km simulation exhibits a high degree of consistency with both the aircraft partial column (Figure 4) and surface sites (Figure 8). We add the following lines in the conclusion for a better explanation:*

"Despite the skill of the 12 km simulations in representing the Pandora column measurements, there appears to be greater hourly variation in the simulation, the aircraft measurements, and the surface measurements than in the Pandora observations. Future work should continue to understand this relationship."

2. Mostly, I can understand what the authors want to say. However, I suggest further improvement of the language of the manuscript. Please find below for further details.

*Ans: We thank the careful examination of language and have addressed all of the corresponding suggestions below.*

Minor comments:

Line 22: "campaign" to "campaigns"

*Ans: Corrected.*

Line 24: "Pandora columns" to "Pandora NO2 columns"? In addition, do you refer to vertical columns or slant columns? Please use accurate terms.

*Ans: Corrected.*

Line 25-26: Please rewrite the second part of the sentence. I understand what you want to say, but the sentence needs to be clarified.

*Ans: Clarified as follows in lines 25-26:*

"hourly variation in the column effective temperature driven by the fractional contribution of atmospheric layers to the total $NO_2$ column"

Line 26: Again, what are the Pandora observations? NO2 vertical columns? Tropospheric or total?

*Ans: Total $NO_2$ vertical columns. Corrected in main text.*

Line 31: "versus" to "against" and delete the second "versus".

*Ans: Corrected.*

Line 37: "at the surface" to "NO2 surface concentrations"?

*Ans: Corrected.*

Line 39: What do you mean by the integral of weakly connected layers? The sentence following it? Please rewrite the sentence which is too long.

*Ans: Corrected as follows in lines 39-40:*

"the differences in hourly variation of atmospheric layers; with the lowest 500 m exhibiting greater $NO_2$ concentrations in morning and evening"

Line 48: Add "spatial" before "gaps".

*Ans: Corrected.*

Figure A1 (Line 670): Add the period of the DISCOVER-AQ Colorado campaign.

*Ans: Corrected.*

Line 136-137: Please rewrite the sentence.

*Ans: Re-written in lines 135-136.*

"We obtained the level 2 data product from the version rnvs3p1-8 for PGN and for DISCOVER-AQ (data source listed in the code and data availability section)."

Lines 145-147: Doesn't it depend on the accuracy of the stratospheric NO2 vertical columns?

*Ans: Indeed, we agree with the reviewer, that there is a lack of consistent and accurate hourly stratospheric $NO_2$ observations.*

Line 161: Write down the full name of VOC at its first appearance.

*Ans: Corrected.*

Line 200-202: Please rewrite the sentence and explain Equation (1).

*Ans: Re-written and explained in lines 198 - 202.*

To account for the hourly changes in vertical variation of column temperature, we calculate simulated NO2 effective temperatures $T_{eff}$ using the site-specific hourly GEOS-FP temperature profiles $(T)_i$, NO2 cross section $\sigma(NO_2)_i$, and GCHP NO2 vertical profiles $VC(NO_2)_i$ following equation (1) of Herman et al. (2009):

$$T_{eff} = \frac{\sum_i^N (\sigma(NO_2)_i \cdot VC(NO_2)_i \cdot (T)_i))}{\sum_i^N (\sigma(NO_2)_i \cdot VC(NO_2)_i))} \tag{1}$$

Line 209: Add a period after "photometer".

*Ans: Corrected.*

Figure 2. The observational lines are messed up. Does different coloring make the figures clearer?

*Ans: We altered the figure graphics for better visibility.*

Line 236: How did you calculate normalized biases?

*Ans: We have added section 2.8 that includes equation 3 and the following lines to describe the calculation of normalized biases, in the main text.*

We use normalized absolute bias or normalized bias (NB) to evaluate the simulations. The NB is calculated using the following equation-

$$NB = \frac{\sum_{i=1}^{N}|S_i - O_i|}{\sum_{i=1}^{N} O_i} \times 100\% \tag{2}$$

where, $O_i$ is the observation and $S_i$ is the corresponding simulated value, $i$ refers to the index of the observation and $N$ refers to the total number of observations.

Line 241: What do you mean by left panel?

*Ans: Noted and corrected. We deleted the redundant "(left panel)" in Line 241*

Lines 254-256: Please explain it.

*Ans: Explanation added to the main text.*

The GCHP simulated effective temperature is also warmer for Asian sites, however the effective temperature is lower during the early afternoon when near-surface $NO_2$ concentrations tend to be minimum such that the stratospheric $NO_2$ that makes a larger fractional contribution to the total column.

Equation 2: Why did you multiply 0.2? How did you determine this value?

*Ans: We clarified the scaling factors in lines 267-274*

The corresponding correction factor (CF) for hourly variation in the effective temperature is calculated as:

$$CF = 1 + \left(\frac{1}{0.8} - 1\right) \times \frac{(T_{eff}(GCHP(hour)) - T_{eff}(Pandora(hour)))}{294 - 220} \tag{3}$$

The factor of $\left(\frac{1}{0.8} - 1\right)$ reflects the difference between the $NO_2$ columns fitted with a 220 K $NO_2$ spectrum that are about 80% of those fitted with a 294 K $NO_2$ spectrum.

Lines 278-282: I wonder whether comparing the total NO2 columns in the left panel to partial NO2 columns between 300 m and 4 km in the right panel is correct, although it seems the NO2 columns between 300 m and 4 km dominate the NO2 total columns. In addition, is the left panel of Figure 4 for DISCOVER-AQ Pandora or PGN Pandora?

*Ans: The left panel describes the hourly variation of the total columns (simulated and Pandora observed) as compared to the right panel that describes the partial column (simulated and aircraft) when aircraft observations were available between 300m and 4 km above ground level.*

*We have added the "DISCOVER-AQ" Pandora in the figure caption for better clarification and replaced the term PGN with "Pandora" in the legend.*

Line 282-287: I think you have applied the correction to the corrected Pandora NO2 total columns. If so, how much does it affect the NO2 total columns?

*Ans: We applied the correction to the raw Pandora columns to obtain the corrected Pandora columns.*

*The correction results are described in sec 3.3:*

*Lines 397-400 :*

*The site-specific effective temperature correction factors typically increase the hourly variation of the Pandora observed columns over DISCOVER_AQ sites (3.5% from the daytime mean) and PGN sites (4% from the daytime mean)*

Line 348-349: Please rewrite this sentence.

*Ans: Noted and corrected in lines 362-364 -*

"The middle and right panels show the hourly variation of observed and 12 km simulated surface $NO_2$ mixing ratios averaged over the PGN sites with and without PBLH modification respectively."

Line 397: Does the PBLH modification increase NBs?

*Ans: This is described in sec 3.3:*

Line 302:

"Excluding the PBLH modification would have increased the NB to 13%."

Lines 315-317:

"Excluding the PBLH correction would have increased the NB (eastern US: 12%, western US: 18%, Europe: 18%, and eastern Asia: 26%) with the largest change in Asia."

Line 403: Add "total" before "NO2"?

*Ans: Noted and added.*

Line 408: "versus" to "against".

*Ans: Noted and corrected.*

---

## Author Comment (AC2)

The questions and comments from reviewers are in black, the responses are in black italics and the lines included in the main text are in blue.

**Reviewer 2**

General comment This study explores the diurnal variation of total NO2 columns and surface NO2 concentrations using GCHP model simulations with independent NO2 measurements from DISCOVER-AQ campaign and Pandonia Global Network. Two corrections are applied on PGN/Pandora total NO2 columns to better represent the dependence of NO2 cross section on the temperature, and different local solar time along the PGN/Pandora line-of-sight. Besides, the authors also test the influences of horizontal resolution and planetary boundary layer height (PBLH) modification on the model performance against aircraft and ground-based observations. It is demonstrated that compared with the other two sensitivity runs, fine scale (~12km) modelled NO2 columns with PBLH modification show smaller bias to independent measurements and better agreement in terms of NO2 diurnal variation. Based on model simulations, the authors find that NO2 columns below 500m show much stronger diurnal variation that that of total columns, which is dampened by residual columns above with much weaker variability. The findings of this study are important for understanding the relationship between NO2 columns and surface concentrations, and I recommend it to be published after addressing following issues.

*Response: We thank the reviewer for acknowledging the importance of this work. Specific responses are listed below:*

Specific comments

Line 39: what is "connected layers"?

*Ans: We have modified it to –*

"the differences in hourly variation of atmospheric layers"

Line 200-203: please re-write this sentence and explain the meaning of each term in this equation.

*Ans: Corrected. Lines 198-202:*

To account for the hourly changes in vertical variation of column temperature, we calculate simulated NO$_2$ effective temperatures $T_{eff}$ using the site-specific hourly GEOS-FP temperature profiles $(T)_i$, NO$_2$ cross section $\sigma(NO_2)_i$, and GCHP NO$_2$ vertical profiles $VC(NO_2)_i$ following equation (1) of Herman et al. (2009):

$$T_{eff} = \frac{\sum_i^N (\sigma(NO_2)_i \cdot VC(NO_2)_i \cdot (T)_i))}{\sum_i^N (\sigma(NO_2)_i \cdot VC(NO_2)_i))} \tag{1}$$

Line 241: what do you mean by "left panel" and in which figure?

*Ans: Noted and corrected. We deleted the redundant "(left panel)" in Line 241.*

Line 254-255: why the simulated effective temperature is lowest in the early afternoon? Line 258-260: please explain the scientific meaning of "0.2" and "(294-220)" in the equation.

*Ans: We explain the effective temperature as a function of hourly variation of GHCP simulated total $NO_2$ columns, which show lower concentrations during the early afternoon and increases during the evening. We modified the main text for better clarification.*

*Lines 262 – 265:*

The GCHP simulated effective temperature is also warmer for Asian sites, however the effective temperature is lower during the early afternoon when near-surface $NO_2$ concentrations tend to be minimum such that the stratospheric $NO_2$ that makes a larger fractional contribution to the total column.

*We clarified the scaling factors in lines 267-274:*

The corresponding correction factor (CF) for hourly variation in the effective temperature is calculated as:

$$CF = 1 + \left(\frac{1}{0.8} - 1\right) \times \frac{(T_{eff}(GCHP(hour)) - T_{eff}(Pandora(hour)))}{294 - 220} \tag{3}$$

The factor of $\left(\frac{1}{0.8} - 1\right)$ reflects the difference between the $NO_2$ columns fitted with a 220 K $NO_2$ spectrum that are about 80% of those fitted with a 294 K $NO_2$ spectrum.

Line 289: it seems that PBLH modification has larger impact on simulated NO2 columns in the morning and evening than midday. What is the reason for this?

*Ans: The PBHL modification reduces the NB by 3-4% for total $NO_2$ vertical columns averaged between 9 AM- 6 PM local solar time across all sites. The difference between the impact of PBLH modification on morning/evening columns as compared to midday columns is about 2-3%. This small difference is primarily driven by the increment in $NO_2/NO_x$ ratios in the total column is stronger during the morning and evening, which is further enhanced by the PBLH modification as observed in Figure A6.*

Line 292: in Figure 4 and 5, both corrected PGN/Pandora NO2 columns and aircraft partial NO2 columns present a distinct increase in early morning, which is also found in GEMS NO2 observations. However, this feature is not reproduced in modelled NO2 columns even though NOx emissions have a morning peak around 9:00 a.m. local time. What is the explanation for this discrepancy?

*Ans: The simulated $NO_2$ columns show a peak at 9 AM local time (capturing the morning peak in $NO_x$ emissions) and then consistently reduce till noon. The Pandora and aircraft columns show a late increase (between 10-11 AM local time) that is not well captured by the model. This difference is most likely driven by two factors –*

1. *Even at 12 km, the representation of $NO_x$ emissions peak in the morning could be diluted especially in developed regions where $NO_x$ has been massively reduced so that background $NO_2$ is significant. In Asian sites the model capture the increase because the NOx emissions there are still very strong and distinguished from background $NO_2$.*
2. *The use of coarse meteorological fields.*

*Although we use fine scale emissions and meteorology, we recognize the importance of more detailed emissions and metrological fields. We add lines 423 -438 for better clarification –*

"Despite the skill of the 12 km simulations in representing the Pandora column measurements, there appears to be greater hourly variation in the simulation, the aircraft measurements, and the surface measurements than in the Pandora observations. Future work should continue to understand this relationship. Future work should also leverage the information developed here to test the performance of surface $NO_2$ concentrations inferred from the geostationary constellation against ground-based measurements."

Technical comments

Line 31-35: please simplify this sentence.

*Ans: Done.*

Line 36: change "column" to "columns".

*Ans: Corrected.*

Line 70: please expand the abbreviation "KORUS-AQ" when it appears for the first time. Line 85: please expand the abbreviation "CTMs" when it appears for the first time.

*Ans: Corrected.*

Line 97-102: please combine these two sentences to make it less redundant.

*Ans: Corrected.*

Line 108: please expand the abbreviation "CONUS" when it appears for the first time. Line 136-137: please re-write this sentence.

*Ans: Corrected.*

Line 143: please keep consistent expression of longitude (sign) in Tables A1 and A2.

*Ans: Corrected.*

Line 198: change "identifies" to "identified".

*Ans: Corrected.*

Line 315: change "3.3" to "3.4".

*Ans: Corrected.*

 Line 340: change "3.4" to "3.5".

*Ans: Corrected.*

Line 355: change "3.5" to "3.6".

*Ans: Corrected.*

Line 397: remove the comma.

*Ans: Corrected.*

---

## Author Response (AR2)

**Response**

We thank the editorial team and reviewers for acknowledging the importance of this work and improving the quality of the manuscript.

We thank the editor for suggesting the changes to optimize the abstract and the conclusion. We have incorporated those changes based on the ACP guidelines.

We thank reviewer 1 for suggesting the technical changes. We have corrected them in the final manuscript.